# Impact of Sustainable Soil Cropping Management on the Production and Stability of Bioactive Compounds in *Tanacetum balsamita* L. by Cold Pressure Extraction

**DOI:** 10.3390/plants14060948

**Published:** 2025-03-18

**Authors:** Alessandra Bonetti, Martina Grattacaso, Sara Di Lonardo, Luigi Paolo D’Acqui

**Affiliations:** 1Research Institute on Terrestrial Ecosystems (IRET), National Research Council of Italy (CNR), Via Madonna del Piano 10, Sesto Fiorentino, 50019 Florence, Italy; alessandra.bonetti@cnr.it (A.B.); sara.dilonardo@cnr.it (S.D.L.); luigipaolo.dacqui@cnr.it (L.P.D.); 2National Biodiversity Future Center (NBFC), Piazza Marina 61, 90133 Palermo, Italy

**Keywords:** *Tanacetum balsamita* L., bioinoculants, compost, chlorogenic acid, quercetin

## Abstract

This study investigated the effects of agronomic amendments, such as compost and bioinoculants (mycorrhizal fungi and plant growth-promoting bacteria), and eco-friendly agronomic practices, on the crop yield and stability of extracted *Tanacetum balsamita* L.’s bioactive compounds, which were obtained through cold pressure (Timatic). Plants were cultivated under four treatments: compost, bioinoculant, combination (bioinoculants + compost), and control treatments. After harvesting, the bioactive compounds were extracted and stored for one year at 4 °C and room temperature. Total polyphenols, IC50 values (DPPH test), and anti-radical activity (ORAC test) were assessed, and High-Pressure Liquid Chromatography analyses of polyphenolic profiles were performed. After 12 months, the quantified bioactive compounds exhibited a reduction of 1.5% to 5.6% at 4 °C, while more pronounced decreases were observed at room temperature: control (93%), compost (8.9%), bioinoculant (32.7%), and bioinoculant + compost (93.4%). Moreover, antioxidant and anti-radical activity were maintained in all treatments at 4 °C, whereas only the bioinoculant and compost treatments exhibited these activities at room temperature. The analysis of bioactive compounds in the 4 °C extracts indicated a statistically significant decline in quercetin and chlorogenic acid across all treatments, with caffeic acid remaining detectable after 12 months. In contrast, at room temperature, chlorogenic acid, di-caffeoylquinic acid and quercetin were retained only in the bioinoculant and compost treatments.

## 1. Introduction

The pursuit of sustainable agricultural practices that can deliver yields similar to those of high-intensity farming is essential for tackling the challenges of climate change, while simultaneously reducing the environmental impact [1]. Following the green revolution, there is an emerging need for a ‘microbial revolution’, which focuses on utilizing and manipulating plant microbiota as sustainable tools to enhance plant productivity [2,3,4]. In recent years, crop production has experienced significant advancements, largely due to innovative technologies, including integrated nutrient management practices that employ biofertilizers. These biofertilizers encompass phosphate-solubilizing bacteria (PSBs), both symbiotic and non-symbiotic N_2_-fixing bacteria, as well arbuscular mycorrhizal (AM) fungi. The increasing adoption of biofertilizers to enhance plant growth and yield has gained momentum in recent years, driven by the rising costs and environmental hazards associated with chemical fertilizers [5]. Research has demonstrated that N2-fixing bacteria and arbuscular mycorrhizal fungi can significantly improve the growth and production of various fruit plants [6,7]. These beneficial microorganisms also improve microbiological activity in the rhizosphere [8]. The enhanced nutritional status of plants induced by these microorganisms has been linked to an increase in the content of valuable secondary metabolites and overall plant production in several crops, including *Zea mays* L., *Solanum lycopersicum* L., *Capsicum annuum* L., *Ocimum basilicum* L., *Mentha* spp., *Echinacea purpurea* (L.) Moench., *Artemisia annua* L., *Stevia rebaudiana* (Bertoni) Bertoni, *Allium sativum* L., and *Hypericum perforatum* L. [2,9,10,11,12,13].

Organic fertilizers or organic amendments, such as green manures, biodigestates, and composts, can play a key role in maintaining or enhancing soil fertility, while at the same time contributing to climate change mitigation [14]. Compost benefits soil in various ways, such as reducing the bulk density and improving the water-holding capacity, as well as exerting an antagonistic effect on pests. The incorporation of compost into soil enhances microbial activities that, in turn, promote the mineralization and recycling of organic substances, finally leading to increased crop productivity [15].

Jin et al. [16] reported that strawberries cultivated using organic methods exhibited higher levels of antioxidant capacities, polyphenols, and flavonoids compared to those grown conventionally. Notably, compost not only enhanced the productivity of the crops but also enriched the polyphenol content and antioxidant capacity in *Mesembryanthemum edule* [17].

Costmary (*Tanacetum balsamita* L.) (syn. *Chrysanthemum balsamita* L., *Balsamita major* Desf.) is a medicinal plan, belonging to the Asteraceae family. It is a large perennial plant of Asian origin with yellow flowers, and it has been grown in Europe and Asia since the Middle Ages [18]. In Italy, *Tanacetum balsamita* L. has been known since the Greek and Roman age, but the first botanical description was found during the IX century A.D. [19]. Many publications focused on particular components of *T*. *balsamita*. Baranauskienė et al. [20] identified five compounds in the o-dihydroxyphenolic fraction of *T. balsamita* var. *tanacetoıdes*, which included caffeic, chlorogenic, and ferulic acids. Todorova and Ognyanov [21] isolated seven germacranolides from the flowers of a *T. balsamita* population cultivated in Bulgaria and found these compounds were different from the presently known eudesmanolides in a *T. balsamita* population cultivated in Poland. Recent research on *Tanacetum balsamita* L. plants cultivated in Tuscany, Italy, has found chlorogenic acid, di-caffeoylquinic acid, and quercetin to be the main polyphenols and flavonoids present in extracts from leaves of this species [22].

In the XVII century, an herbal preparation containing the essential oil of *T. balsamita*, cinnamon, and other spices (*Acqua Antisterica* in Italian, i.e., anti-hysterical water) was developed in Tuscany, Italy, and rapidly gained popularity in Europe [19]. Today, in Tuscany, cold pressure extraction is mainly employed to obtain aqueous extracts of this species for uses in cosmetic preparations [19,23]. The use of water instead of a solvent for extraction renders these preparations safe and suitable to be used in cosmetical and food products.

The main objective of this study was to assess the effects of bioinoculants and compost on the stability of aqueous plant extracts of the medicinal plant *Tanacetum balsamita* L., obtained by TIMATIC extractions. The employment of antioxidant and anti-radical activity tests and polyphenol characterization permitted us to assess the efficiency of the extracts. The combined use of TIMATIC and sustainable agronomical practices could be exploited for the use in cosmetic, pharmaceutical, and food preparations.

## 2. Results and Discussion

### 2.1. Total Polyphenols

The concentrations of total polyphenols in aqueous extracts of *Tanacetum balsamita* L. under different treatments are shown in Table 1. In June 2023, bioinoculant (B) and compost (CP) treatments exhibited the highest concentrations, measuring 518.43 ± 9.41 mg/kg and 567.63 ± 8.44 mg/kg, respectively, while in June 2024, a slight increase in total polyphenols was observed for samples held at 4 °C. However, among the samples at room temperature (RT), only the B and CP treatments showed detectable levels of total polyphenols after 12 months. The Folin–Ciocâlteu (F-C) method is a commonly used technique for assessing the total phenolic content due to its simplicity and reproducibility. However, it is important to note that this method is not specifically designed for specific phenolic compounds, as the reagent could be reduced by other non-phenolic compounds present in the sample, potentially leading to the overestimation of the total polyphenol content [24]. This is the case with the higher concentration of TP measured in June 2024 after 12 months at 4 °C, where caffeic acid (CA), obtained by chlorogenic acid (CGA) degradation, was detected by the F-C test (see Section 3.2).

### 2.2. HPLC Analysis

The composition of aqueous extracts of *Tanacetum balsamita* L. at the time of harvest and after 12 months, either at 4 °C or RT, was assessed by Diode Array High-Pressure Liquid Chromatography (DAD-HPLC). At time 0 (T0), for each treatment, three main compounds were identified: chlorogenic acid (CGA), with a retention time (*rt*) of 12.43 min, di-caffeoylquinic acid (DCQ), with a *rt* of 13.62 min, often found together with CGA [25], and quercetin (QT), with a *rt* of 18.47 min (Figure 1).

In particular, QT’s concentration at T0 resulted in being positively influenced by the CP treatment (Figure 2).

Peaks observed between 15 min and 20 min were identified as flavonoid derivatives.

To our knowledge, this is the first identification of aqueous extracts from costmary leaves using the TIMATIC technique. The composition of these extracts is in accordance with findings from *Tanacetum balsamita* L. extracts with methanol or ethanol [26,27]. The concentrations of CGA, DCQ, and QT accounted for more than 45% of total polyphenols, as determined by the Folin–Ciocâlteau test. After 3 months of storage at 4 °C, both the CGA concentration and QT exhibited a decrease, ranging from 8.8% (bioinoculant + compost (B + CP treatment) to 24.3% (CP treatment), and from 4% (B + CP treatment) to 14.6% (CP treatment), respectively (Figure 3).

After 12 months at 4 °C, all treatments exhibited a decrease in the CGA concentration, to around 50%, with the simultaneous presence of CA (Figure 3 and Figure 4).

Additionally, the flavonoid derivatives detected between 15 and 20 min were found to be completely degraded. The degradation of CGA to CA has been reported in previous studies [28,29]. An increase in DCQ, probably because of the CGA degradation, was also observed. Finally, QT concentrations decreased from 68.6% (B + CP treatment) to 71.6% (CP treatment) by the end of the experiment (Figure 5).

At RT, after 3 months, both CGA and QT remained detectable across all treatments, with concentrations declining between 14.4% (B + CP treatment) and 22.1% (CP treatment) (Figure 5). After 12 months, treatments C and B + CP resulted in the complete degradation of the extract’s compounds (Figure 5). On the contrary, for B and CP treatments, CGA concentrations exhibited a decrease between 60.4% and 57.4%, and QT concentrations exhibited a decrease between 43.4% and 65.4%, respectively (Figure 5). The presence of CA was also observed in B and CP treatments (Figure 4). Finally, the concentrations of CA after 12 months at both 4 °C and at RT were found to be comparable in the B and CP treatments (Figure 6a,b).

### 2.3. Antioxidant Activity (DPPH Test)

The antioxidant properties of the extracts were evaluated using the DPPH test (Table 1). This test exploits a single-electron transfer reaction; it measures the antioxidant reducing capacity and permits us to evaluate the anti-radical activity of studied samples. This test compares the scavenging capacity (expressed as a percentage) and concentration of the sample, permitting a correlation between these two terms. The DPPH test of the T0 aqueous extracts of *Tanacetum balsamita* L. in 2023 exhibited an inhibitory concentration, at 50% (IC50) values of less than 4 µg/mL for all treatments, with slightly but not significantly better values for the B, CP, and B + CP treatments. After 12 months of storage at 4 °C, a decrease in IC50 activity was observed across all treatments, indicating that the higher sample concentration required to the obtain IC50 value, the lower the antioxidant activity (Table 1), with a mean activity of 5.17 ± 1.57 µg/mL. The HPLC analysis suggests that antioxidant activity may be maintained by the presence of QT and CA, resulting from the degradation of CGA. At RT, only the combination of both B and CP treatments (B+ CP) showed an antioxidant activity of 3.69 ± 1.62 µg/mL and 1.70 ± 0.26 µg/mL, respectively (Table 1). Even at RT, the simultaneous presence of QT, CA, and partially degraded CGA could explain the high antioxidant values observed from the DPPH test.

A correlation analysis was performed between IC50 and ORAC values, and extract components (Table 2). The R^2^ values were derived by correlating IC50 and ORAC measurements with the respective concentrations of individual compounds at T0, T3 (3 months), and T12 months post-harvest. Strong correlations were found between QT’s concentration and IC50 values under 4 °C storage conditions after 12 months, with R^2^ values ranging from 1.00, except for the C treatment (R^2^ = 0.45). As well, DCQ exhibited higher correlation coefficients than CGA values, particularly for the B and CP treatments at 4 °C, with R^2^ values of 0.96 and 0.91, respectively. Moreover, DCQ in the C treatment displayed higher correlation indices than CGA and QT (Table 2). These findings are consistent with the work of Pakulskas et al. [30], who attributed the higher IC50 values in di-caffeoylquinic acid to the presence of two caffeic-acid moieties compared to chlorogenic acid.

At RT after 12 months, stronger correlations were found for QT in the CP and B treatments than for CGA and DCQ, with the coefficients of determination R^2^ = 1.00 and 0.99 (Table 2), respectively. In the other treatments, i.e., C and B + CP, as above reported, QT, CGA, and DCQ were degraded. Benedec et al. [31] reported similarly high values for QT in the DPPH test with *Chrysanthemum balsamita* var. *balsamita*. By comparing the coefficients of determination, it can be inferred that phenolic and flavonoid compounds play a significant role in the antioxidant activity of the selected plant extracts. The high correlation values in the IC50 for QT, at 4 °C and at RT, would confirm that the DPPH test is mainly influenced by QT concentrations, while CGA undergoes degradation in the presence of CA and DCQ at higher concentrations. The strong correlation between the DPPH test and QT concentrations in B, CP, and CP + B treatment at 4 °C, as well as in B and CP at RT, could be attributed to the better growing conditions for costmary plants provided by the applied treatments that, in turn, favored QT activity. The overall health of costmary plants could help to prevent the rapid degradation of QT, as evidenced by the results from C treatments at both 4 °C and RT conditions.

### 2.4. Anti-Radical Activity (Oxygen Radical Absorbance Capacity (ORAC) Test)

The anti-radical properties of the extracts were analyzed using the Oxygen Radical Absorbance Capacity (ORAC) test (Table 1). This assay consists in a hydrogen-atom transfer reaction that quantifies the hydrogen-atom donor capacity and provides a measure of oxidative degradation. This test is widely employed to evaluate oxidative degradation in biological samples, dietary supplements, and food products. The ORAC test results of aqueous extracts of *Tanacetum balsamita* L. harvested in June 2023 displayed the highest ORAC activity for B and CP treatments, with values of 8720.12 and 8116.40 µMTE/L, respectively (Table 1). After 12 months at RT, only B and CP treatments showed ORAC activity (Table 1). The correlations between ORAC values and the concentration in costmary’s extracts of QT, CGA, and DCQ are shown in Table 2. At 4 °C, the CP treatment exhibited high correlation values for CGA (R^2^ = 0.83), while the CP + B treatment exhibited high correlation values for QT and CGA (R^2^ = 0.81 and 0.97, respectively). At RT, the B treatment exhibited the best correlation for QT (R^2^ =0.99). (Table 2). Despite the fact that CGA is reported as a strong hydroxyl- and superoxide scavenger, in which the hydroxyl group from phenolic acid donates a hydrogen atom to free radicals [32], its correlation value was R^2^ = 0.67. This relatively lower correlation may be attributed to the degradation of CGA into CA.

## 3. Materials and Methods

### 3.1. Experimental Plots

*Tanacetum balsamita* L. plants (540) were collected by a nursery at the five-leaves stage and transplanted in “Giardino Santa Maria Novella” of “Officina Profumo Farmaceutica Santa Maria Novella” near Florence.

A complete randomized design in a set of 12 replicates (3 for each treatment) was used to conduct the experiment. The urban compost was from organic waste (Sienaambiente, Italy) (Table 1) and was applied at 60 q/ha, and the bioinoculant was from Bioseed—Unmaco Company (Novara, Italy), contained mycorrhizae 5% (*Rhizoglomus irregulare*, *Glomus mosseae*, *Funnelliformis caledonium*) and rhizosphere bacteria at 10^10^ u.f.c./g (*Azotobacter vinelandii*, *Rhodopseudomonas palustris*, *Bacillus megaterium*), and was sprayed at a 2‰ rate. An organic liquid fertilizer at a 2‰ rate for helping microbial activity from Nutribio N—Unmaco (commercial) was added.

Four treatments were made: (1) control (C), consisting of any compost and/or bioinoculant; (2) compost (CP), which was added at 60 q/ha; (3) bioinoculant (B), which was added at a 2‰ rate; and (4) bioinoculant + compost (B + CP+), which was the sum of the two treatments. Each plot consisted of 45 plants in 5 rows with 9 plants in each row. From each plot, at the end of the experiment, 27 central plants were assayed, avoiding plants on the border.

Plants were transplanted in February 2023, treatments were applied in April 2023, and sampling was performed in June 2023, when plants reached “balsamic time”, which is the period with the highest concentration of polyphenols.

### 3.2. Timatic Extraction

Leaves collected at the “balsamic time” from each treatment were processed at the “Officina Profumo Santa Maria Novella”, in Florence. The extraction of leaves in water was performed using a Timatic Instrument (Tecnolab, Spello, Perugia, Italy). Briefly, 1.6 kg of *Tanacetum* leaves from each treatment (3 plots for each treatment) were processed at 30 °C in Timatic with 9.6 L of water and citric acid at 0.6% (*w*/*v*) for a total of 40 cycles, with a pressure between 6.5 and 9.0 bar. Final extracts (5L) were conserved in 5L aluminum tanks at 4 °C and 88% of relative humidity, until analysis. Aliquots of extracts were analyzed after sampling in 2023 (T0), and the same aliquots were placed at 4 °C and at room temperature (RT) and analyzed 3 months (T3) and 12 months (T12) after harvest.

### 3.3. Chemical Analysis

#### 3.3.1. Total Polyphenols Quantification

The total phenol assay was performed by using the Folin–Ciocâlteau (F-C) reagent, as described by Biagi et al. [33]. A total of 500 µL of the aqueous extracted samples were placed in 25 mL flasks, where 2.5 mL of the F-C reagent was added, and the mixture was shaken for 30 s. Afterwards, 5 mL of a saturated Na_2_CO_3_ water solution was added and the mixture was brought to volume. The absorbance of the colored reaction product was read at 730 nm using a UV-Visible spectrophotometer Cary 50 Scan (Varian, Palo Alto, CA, USA) and using distilled water as the blank. Results were expressed as mg of Gallic Acid Equivalent for g of dry weight (mg GAE/g DW). The total phenolic compound extractions and quantification were performed in triplicate.

#### 3.3.2. HPLC Analysis

Analyses of aqueous *T. balsamita* L. extracts were carried out using a Thermo Finnigan supplied with an UV-Vis Diode Array Detector Spectra System UV6000 LP (ThermoQuest Corporation, Austin, TX, USA) a binary pump Spectra System P4000. Chromatograms were collected and analyzed by using ChromQuest Software (1998, v. 2.51, ThermoQuest Corporation, Austin, TX, USA). For the separations, a Kinetex Phenyl-Hexyl 100 A (Phenomenex, Torrance, CA, USA), 150 × 4.6 mm reverse-phase C18 column with an identical pre-column, operating at 25 °C, was employed. The eluent was composed of (A) H_2_O/H_3_COOH (99.9:0.1) and (B) MeOH/H_3_COOH (99.9:0.1). A three-step linear solvent gradient system was used, starting from 5% to 99% of solution B, for a 33 min period at a flow rate of 0.8 mL/min. The injection was 15 µL. UV–Vis spectra were recorded in the 220–700 nm range and the chromatograms were recorded at 330 and 350 nm. Analyses at 4 °C and at RT were performed at T0 (right after harvesting), T3, and T12.

#### 3.3.3. Standard Solution, Calibration Curves, and Calculation of Hydroxycinnamic Acids and Flavonoid Content

Chlorogenic acid (5-O-caffeoylquinic acid) (CGA), di-caffeoylquinic acid (4-trans-O-caffeoylquinic acid) (DCQ), and quercetin (QT) (Merck, Darmstadt, Germany) standards were used to confirm the identity of compounds and to quantify them. The stock solutions were prepared by dissolving the crystalline standard in 1000 ppm in 100% methanol. Subsequently, the solutions were diluted to 5 ppm with 80% methanol (*v*/*v*). Five-point calibration curves were obtained for each standard with the following linearities: quercetin, R^2^ = 0.99; chlorogenic acid, R^2^ = 0.9361; di-caffeoylquinic acid, R^2^ = 0.99. These curves were obtained by plotting the standard concentrations as a function of the peak area obtained from HPLC analyses. For this purpose, the stock solutions of the standards were diluted with 80% methanol to 5 different concentrations, ranging from 10 to 320 µg/mL. Each concentration was analyzed in triplicate. Retention times were as follows: chlorogenic acid, *rt* = 12.20 min; di-caffeoylquinic acid, *rt* = 14.52; and quercetin, *rt* = 18.57.

### 3.4. Antioxidant Activity (DPPH Assay)

The free radical scavenging activity using the 1,1 Diphenyl-2-Picryl-Hydrazil (DPPH) reagent was determined for the sample *T. balsamita* L. leaves according to Bonetti et al. [22]. Diluted water extracts of *T. balsamita* L. (1 mL) was added to 1 mL of the freshly prepared ethanolic DPPH solution (0.9 × 10^−4^ M) and stirred. The decolorizing process was recorded at the beginning and after 20 min of reaction at the 517 nm absorbance and compared with a blank control. Antioxidant activity was calculated according to the following formula:Antioxidant activity = [(control absorbance − sample absorbance)/control absorbance] × 100 

Results were expressed as mg/g DW of extracts necessary to inhibit 50% of DPPH. Analyses at 4 °C and RT were performed at T0, T3, and T12.

### 3.5. Oxygen Radical Absorbance Capacity (ORAC) Assay

The method was adapted from the one described by Cao and Prior [34] and the instrument used was a fluorescence spectrophotometer (Varian Cary Eclipse) (Palo Alto, CA, USA). The sample was added to a free-radical generator (AAPH, 2,2′-azobis(2-aminopropane) dihydrochloride) and the inhibition of the free radical was measured. Fluorescein was used as a target for the free-radical attack. Free radicals caused conformational changes in the structure of fluorescein, leading to dose- and time-dependent fluorescence quenching. The following solutions were added to a quartz cuvette: 2738 µL of fluorescein (25.5 mg/L solution, maintained at 4 °C), 37 µL of phosphate buffer solution (75 mM, pH 7.4), and 150 µL of Trolox standard (Sigma-Aldrich, 100 µM), blank (buffer solution), or sample solution. After incubation at 37 °C for 30 min, the addition of 75 µL of the AAPH solution (86.8 mg/mL in buffer solution and kept in ice) started the reaction. The exciting λ was 490 nm and the emission λ was 512 nm. The total antioxidant capacity or ORAC unit (µM) was obtained by the following formula:ORAC unit (μM) = [20 k (S_sample_ − S_blank_)]/(S_Trolox_ − S_blank_)
where k is the dilution factor, S_sample_ is the area under the curve of the sample, S_blank_ is the under-curve area of the blank, and S_Trolox_ is the under-curve area of the standard.

ORAC values were expressed as µM Trolox Equivalents/L of extract using the standard curve established previously. Analyses at 4 °C and RT were performed at T0, T3, and T12. All antioxidant assays were repeated three times.

### 3.6. Statistical Analysis

A one-way analysis of variance (ANOVA) was used to evaluate the levels of statistical significance, tested by a post hoc comparison test (Tukey test HSD) at *p* < 0.05 for all analyses (Statgraphics Plus, version 5.1 for Windows). All data are reported as the mean ± standard deviation of three measures. Linear regression curves were calculated by the Excel program in Windows.

## 4. Conclusions

This study aimed to evaluate the effects of sustainable cropping management on the stability of *Tanacetum balsamita* L. leaves extracts, obtained by Timatic extraction. The extracts, rich in chlorogenic acid, quercetin, and di-caffeoylquinic acid, exhibited a strong antioxidant activity that was highly correlated with the concentration of these single components. The bioinoculant, compost, and combination treatments positively influenced the stability of the extracts, maintaining their concentration and antioxidant activity at 4 °C until 12 months after treatment. Notably, the B and CP treatments were particularly effective at RT for the stability of the extracts, where a high antioxidant activity persisted even after 12 months.

Timatic extraction, performed at 30 °C, effectively prevented the rapid degradation of CGA, as observed by Dawidowicz and Typek at temperatures around 100 °C [29]. Although a partial degradation of CGA was noted in aqueous extracts after 12 months, the residual extract still exhibited relatively high levels of antioxidant and anti-radical activity.

These findings offer significant potential for various applications within the nutraceutical sector, such as food, pharmaceuticals, and cosmetics, promoting the sustainable production of stable and viable nutraceutical compounds.

## Figures and Tables

**Figure 1 plants-14-00948-f001:**
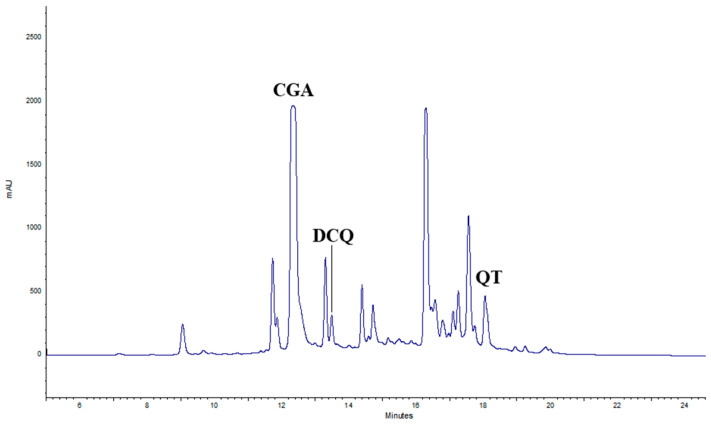
HPLC chromatogram of TIMATIC extracts of June 2023 (T0). CGA = chlorogenic acid, DCQ = di-caffeoylquinic acid, QT = quercetin.

**Figure 2 plants-14-00948-f002:**
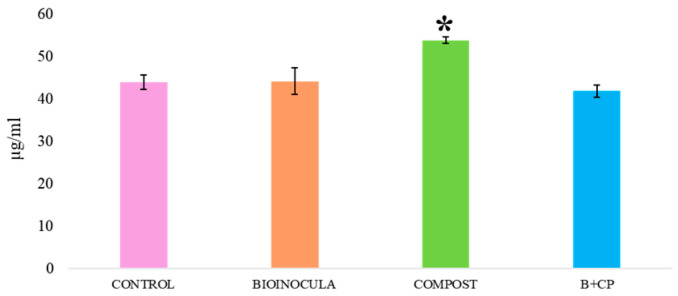
Quercetin concentration in aqueous TIMATIC extracts at T0, according to treatments. (* indicates that the concentration was statistically significant for *p* < 0.05).

**Figure 3 plants-14-00948-f003:**
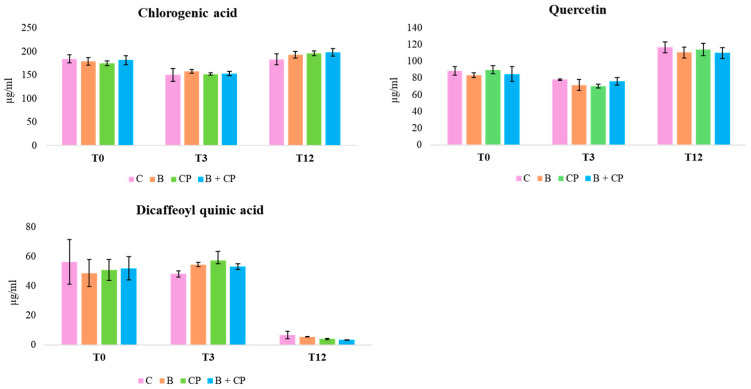
Concentration trends of CGA, DCQ, and QT, during the 12-month period at 4 °C.

**Figure 4 plants-14-00948-f004:**
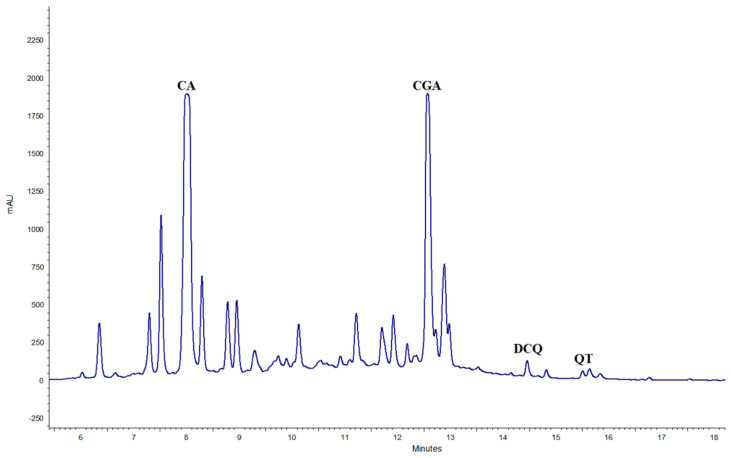
Chromatogram of TIMATIC extract at T12 after 12 months at 4 °C (CA = caffeic acid).

**Figure 5 plants-14-00948-f005:**
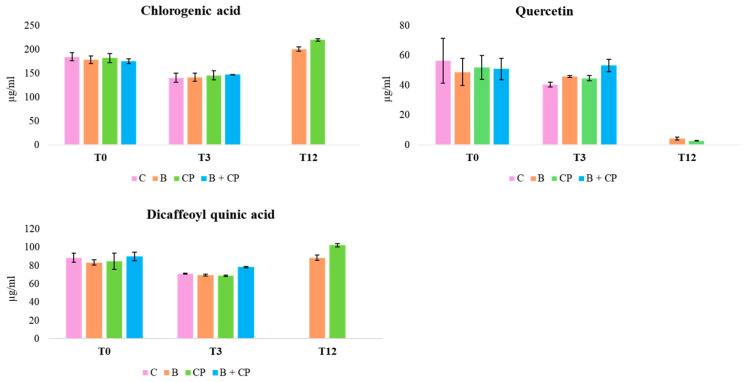
Concentration trends of CGA, DCQ, and QT, during the 12-month period at RT.

**Figure 6 plants-14-00948-f006:**
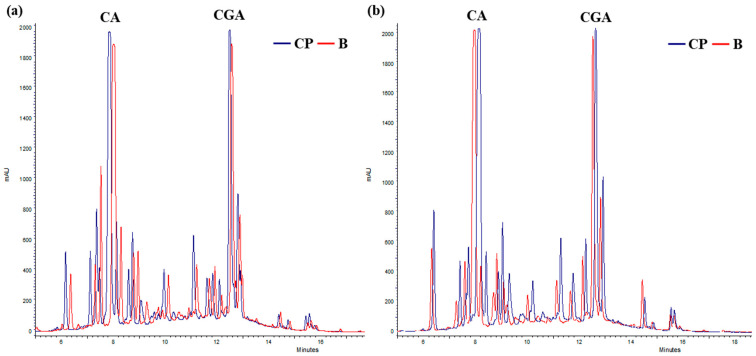
Chromatograms of TIMATIC extracts after 12 months for B and CP treatments. (**a**) Extract held at 4 °C; (**b**) extracts held at RT.

**Table 1 plants-14-00948-t001:** Stability of total polyphenolics (TPs), IC50 (DPPH test), and anti-radical activity (ORAC test) values in aqueous extracts of *Tanacetum balsamita* L. both at 4 °C and room temperature (RT) in the different treatments: compost (CP), bioinoculant (B), a combination of both (B + CP), and control (C). Data were collected in June 2023 and after 12 months (June 2024). Data are expressed as mean ± standard deviation of 3 measures. n.d., values not detected.

	June 2023			June 2024		
	4 °C	4 °C	4 °C	4 °C	4 °C	4 °C
	TP (µg/mL)	IC50 (µg/mL)	ORAC (µMTE/L)	TP (µg/mL)	IC50 (µg/mL)	ORAC (µMTE/L)
C	461.35 ± 22.25	3.15 ± 1.49	5679.41	476.63 ± 0.11	4.72 ± 1.97	10,656.07
B	518.43 ± 9.41	1.10 ± 0.13	8720.12	521.11 ± 0.04	4.45 ± 2.35	n.d.
CP	567.63 ± 8.44	0.87 ± 0.35	8116.40	489.99 ± 0.08	8.42 ± 0.64	8812.67
CP + B	461.11 ± 19.77	0.62 ± 0.30	2845.86	532.11 ± 0.08	3.10 ± 1.31	809.69
	**RT**	**RT**	**RT**	**RT**	**RT**	**RT**
C	461.35 ± 22.25	3.15 ± 1.49	5679.41	31.11 ± 0.05	n.d.	n.d.
B	518.43 ± 9.41	1.10 ± 0.13	8720.12	381.99 ± 0.08	3.69 ± 1.62	21,448.69
CP	567.63 ± 8.44	0.87 ± 0.35	8116.40	516.84 ± 0.41	1.70 ± 0.26	20,196.29
CP + B	461.11 ± 19.77	0.62 ± 0.30	2845.86	30.37 ± 0.02	n.d.	n.d.

**Table 2 plants-14-00948-t002:** Correlation coefficients (R^2^) between IC50 values (DPPH test), anti-radical activity (ORAC test) values, and quercetin (QT), chlorogenic (CGA), and di-caffeoylquinic acid (DCQ) concentrations in aqueous extracts of *Tanacetum balsamita* L. at both 4 °C and RT, as influenced by different treatments: compost (CP), bioinoculant (B), a combination of both (B + CP), and control (C). The R^2^ obtained value is based on the results of 3 analyses conducted at different time points: T0, T3, and T12. n.d., values not detected.

		QT	CGA	DCQ
	DPPH test			
**4 °C**	C	0.45	0.31	0.87
	B	1.00	0.70	0.97
	CP	1.00	0.60	0.91
	CP + B	1.00	0.77	0.8
**RT**	C	n.d.	n.d.	n.d.
	B	0.99	0.65	0.60
	CP	1.00	0.01	0.60
	CP + B	n.d.	n.d.	n.d.
	**ORAC test**			
**4 °C**	C	0.001	0.89	0.71
	B	n.d.	n.d.	n.d.
	CP	0.42	0.83	0.55
	CP + B	0.81	0.97	0.01
**RT**	C	n.d.	n.d.	n.d.
	B	1.00	0.67	0.51
	CP	n.d.	n.d.	n.d.
	CP + B	n.d.	n.d.	n.d.

## Data Availability

The data of this study will be available on request to the corresponding author.

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
