# Peer review of "Impact of Sustainable Soil Cropping Management on the Production and Stability of Bioactive Compounds in Tanacetum balsamita L. by Cold Pressure Extraction"

_plants, 2025, doi:10.3390/plants14060948_

Round 1
Reviewer 1 Report
Comments and Suggestions for Authors
According the authors the manuscript have 3 main objectives: 1. to assess the effects of bioinoculants and compost on the medicinal plant Tanacetum balsamita L., in comparison to traditional agronomical practices, 2. to evaluate the efficiency of cold pressure extraction and 3. to evaluate the stability of aqueous plant extracts. In my opinion, the results obtained were presented in an unorganized manner, making them difficult to read. The order followed by the figures and tables could be reorganized to facilitate reading and understanding of the work. The authors could present the results obtained following the order of the objectives. In my opinion, the objective 1 was little discussed and seems to me to be the most interesting part of the work.
Author Response
Please see the responses in attachment.

Reviewer 2 Report
Comments and Suggestions for Authors
The ms: Impact of Sustainable Soil Cropping Management on the Production and Stability of Bioactive Compounds in Tanacetum balsamita L. by Cold Pressure Extraction, presents interesting results related to the use a non common method for the extraction of natural products from plants. It is well described in all the sections, but there are some details that the authors should be consider to increase its quality. Some of them are:
i. In the results section, page 7 related to the table 1, it is not indicated what is the control in all the variables presented there.
ii. Page 10 line 321 says: ........free radicals caused conformational changes in the protein structure of fluorescein,......what do you mean to protein structure of fluorescein?
If you revise the concepts and terms, fluorescein is not a protein.
iii. Some of the references are not cited correctly as that of ref 23 from Gallori, et al 2001.
The range of pages is wrong, please check such data.
iv. About the analysis by HPLC, it is not described the pattern of the standards, f they were used to have a more precise method of identification of the phenolics detected.
v. Some of the references are so old, like that of Tamas, et al 1989, besides it is not easy to find such reference.
Author Response
Please see the responses in attachment.

Reviewer 3 Report
Comments and Suggestions for Authors
The paper of Bonetti et al. “Impact of Sustainable Soil Cropping Management…” aimed to chemical and bioactivity study of Tanacetum balsamita cultivated under four treatments. Generally, the work contains controversial statements and requires additional material.
Highlights and strengths of the manuscript are:
The obtained results may further increase the interest in Tanacetum balsamita as a source of bioactive compounds.
Specific comments and suggested revisions:
- Timatic extraction procedure does not allow us to understand whether the final volume of extracts was brought to any precise value. So far it looks like the final volume does not matter, which raises doubts about the correctness of the organization of the experiment. It is incorrect to compare samples whose volumes differ.
- Table 1. Part of the table (lower left) repeats its own data, which is not necessary. Why are the data given in μl/ml and not in grams of raw materials? It's completely unclear how to compare them (IC50 calculated as μl/ml – no idea what is that). Check the column designations and add statistic data. I don't understand where the control is. What is “nd”?
- The results of the study of samples after storage are puzzling and appear to be incorrect. If you used normal agricultural procedures without adding anything foreign to the extracts, then the rules of the change in chemical composition for samples under the same conditions should be close. Third line, sample CP (?) – DPPH data changed from 0.87 to 8.42 while CP from first line – from 3.15 to 4.72. ORAC data varied for CP from 5679 to 10656, for 2nd CP from 8116 to 8812, for CP+B from 2845 to 809. There is no rational explanation for such changes other than that they are incorrect. Sorry, but this data cannot be published without serious analysis.
- The quality of the chromatographic analysis leaves much to be desired, since it is obvious that the conditions for optimal separation were chosen incorrectly. The peaks are poorly separated and the presence of rider peaks cannot be guaranteed, as the shape of the peaks is obviously asymmetrical.
- Figure 2 data demonstrated concentration only for quercetin although later data are already given for three compounds. Why?
- It is unclear why CQAs in some samples were destroyed when stored at room temperature, while others were not, although the trend for all samples was maintained when stored at a lower temperature.
- Table 2. Balsamita major. Why? I don't understand what data table 2 is based on. For each sample you have one compound concentration value and one activity value. In a two-dimensional coordinate system, this gives one point. How is the correlation obtained? You can build a correlation for just a group of data, such as all samples stored at room temperature, etc.
- The stated purpose of the work concerns the study of the plant. You devoted the entire text to the study of extracts.
With all due respect to authors, I see no possibility to recommend paper for publication. Authors should clearly formulate the purpose of the work, remove unnecessary material, add correct and reliable data and a clear discussion.
Author Response
Please see the responses in attachment.

Reviewer 4 Report
Comments and Suggestions for Authors
The article is difficult to read because it continually usages abbreviations.
Abstract In the abstract avoid the use of acronyms and avoid abbreviating words.
In the text the first time they are named they should be spelled out in full and then they should be abbreviated.
Results and discussion
Page 3 Line 90-92 Delete the sentence: “This section may be divided by subheadings. It should provide a concise and precise 90 description of the experimental results, their interpretation, as well as the experimental 91 conclusions that can be drawn. ” because of its usefullness.
Page 7 Table 1 Check the CP values, are they correct?.
Pag 7 Line 173 write “a combination of both (CP+B)”.
Page 7 Line 185 “who attributed the higher DPPH values in dicaffeoylquinic acid to the presence of two caffeic-acid moieties with respect to chlorogenic acid” is the phrases correct?
Page 8 Line 208 Why synonimus Balsamita major L. is reported instead of Tanacetum balsamita L.? If you want to use the synonimus Balsamita major you have to cyte the correct author Desf.
Pag 9 Line 237 correct “while C+B treatment exhibited…” with “while CP+B treatment exhibited…”
Pag 9 Line 239 “for QT (R2=0.98)” is it correct? In Table 2 there isn’t a value of 0.98 at this voice.
Pag 9 Line 241 “its correlation value was 0.66.” Is it correct? Because in table there isn’t this value.
3.3.3. Standard solution, calibration curves and calculation of hydroxycinnamic acids and flavonoid content
Pag 10 Well-stated, but could improve by including a table with all HPLC quantitative parameters and statistical analysis data (range of data, F, F critic and recovery).
3.6 Statistical analysis
Well-stated, but could improve by including more description about how the values were obtained.
“One-way analysis of variance (ANOVA) was used to evaluate the levels of statistical significance tested by a post hoc comparison test (Tukey test HSD) at p <0.05 for all analysis (Statgraphics Plus, version 5.1 for Windows). All data are reported as mean of replicates ± standard deviation. Linear regression curves were calculated by Excel program by Windows.”
References Line 450 and 452 delete the highlighted sentence.
Comments on the Quality of English LanguageThe English could be improved to more clearly express the research.
Author Response
Please see the responses in attachments.

Round 2
Reviewer 1 Report
Comments and Suggestions for Authors
The authors have revised the manuscript and most of the suggestions have been addressed. The manuscript is eligible for publication. However, the authors presented the revised version with the changes marked, which makes the document difficult to read. I suggest that the revised manuscript be presented using a different color in the text.
Reviewer 3 Report
Comments and Suggestions for Authors
The authors taking into account the observations made by the reviewer. Considering the explanation provided by the authors the paper after correction may accepted in present from.